# COMPOSING HUMAN-OBJECT INTERACTION WITH DECOUPLED PROTOTYPE FOR ZERO-SHOT LEARNING

## ABSTRACT

Zero-shot Human-Object Interaction (HOI) detection is a daunting problem, largely stemming from the *combinatorial explosion* of potential action-object pairs. Current studies predominantly address this issue by transferring knowledge from large-scale pre-trained models (*e.g.*, CLIP), yet ignore a more straightforward idea, *i.e.*, mimic the powerful compositional generalization ability of human intelligence based on past cases. Besides, they simplify this combinatorial challenge by operating under the assumption that knowledge about unseen compositions is accessible, which is usually impractical in reality. In this work, we extend prior *Closed-World* zero-shot setting to an *Open-World* scenario, where the search space for HOI compositions is entirely unrestricted. For this challenging task, we introduce PROTOHOI, a fresh prototype-based framework for zero-shot HOI detection, which consists of: **i)** distill a set of prototypes from HOI proposal embeddings to model the inherent properties of objects and actions in the context of HOI. **ii)** recalibrate the representation space learned by the HOI detector based on these derived prototypes in a decoupled manner, thereby facilitating the prediction of unseen HOI compositions. Extensive experiments on two standard benchmarks demonstrate the superiority of ProtoHOI over the state-of-the-art methods across all zero-shot settings. The source code will be released.

## 1 INTRODUCTION

Human-Object interaction (HOI) detection is a core human-centric relational detection task that requires identifying all possible combinations of `human action` and `object` within images. Over the past decades, the complexity of HOI recognition appears to be effectively addressed by existing methods [1–3], as demonstrated by their high performance on established benchmarks. However, these HOI detectors are built upon an oversimplified assumption: *they arbitrarily constrain the set of possible HOI relationships, and limit their evaluation to this restricted subset*. For instance, HICO-DET contains 80 object categories and 117 actions, yet provides only 600 valid HOI combinations, while the search space for possible combinations reaches 9,360. This disparity highlights the fundamental challenge of *combinatorial explosion* in HOI detection, which conflicts with the limited pre-defined labels in existing datasets, posing a major obstacle to real-world applications.

To develop practical HOI detection systems, zero-shot HOI detection has garnered significant attention as a critical research direction. Current work [1, 4, 5] commonly partitions labels in existing datasets into *seen* and *unseen* sets based on their visibility during training. However, they usually assume that the unseen compositions are known a *priori* at test time, thereby constraining the search space to a pre-defined set of labels within the dataset, *i.e.*, *Closed-World* zero-shot HOI detection. This paper seeks to refocus the attention of the HOI community on the critical yet long-overlooked *combinatorial explosion* challenge, and propose a more realistic *Open-World* zero-shot scenario that imposes no search space restrictions during testing. Notably, while zero-shot HOI detection also involves generalizing to *unseen objects* [5] or *unseen human verbs* [1], we focus on the compositional zero-shot learning for HOI detection, which requires to learn a well-structured representation space that facilitates the composition of known elements to recognize novel interactions. The key question naturally arises: *how to decouple the HOI representation learning with respect to the inherent properties of objects (e.g., affordance, functionality) and actions (e.g., human poses, gestures)?*

Psychologists reveal that human conceptual representations are decomposable, with compositionality serving as a core component in a human-like learning system [6, 7], exemplified by the capacity of humanoid to recognize novel combinations by using past learned cases [8, 9]. Drawing from this inspiration, we are motivated to lever-

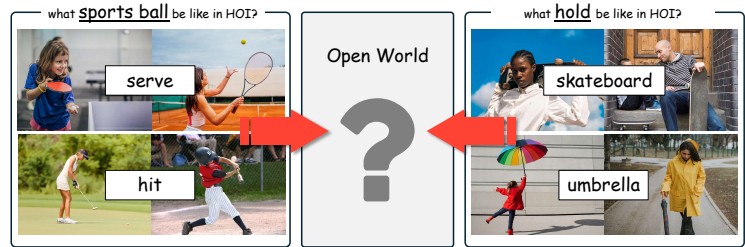

Figure 1: To construct a world, it is essential to comprehend the fundamental nature of its basic constituents; the most straightforward approach to characterize them is through examples.

age prototype learning [10–12] to solve the *combinatorial* problem in zero-shot HOI detection. However, the **challenge ❶** arises from substantial *intra-class diversity* exhibited by actions and objects within the context of HOI. For example, the verb `serve` in the context of `sports ball` may refer to a table tennis or a badminton, while `kick` typically associates with soccer. Similarly, the action `open` entails distinct postures and hand gestures, when applied to different objects such as a `book` versus an `umbrella`. Even within defined HOI categories, behavioral manifestations exhibit contextual variability across scenes. These variations make it difficult to represent HOI triplets using a unified prototype. The **challenge ❷** stems from the *vast number of proposals* generated by HOI detector, which makes it difficult to distill representative prototypes from highly redundant candidate instances. Moreover, HOI detectors typically comprise complex network structures architectures composed of *distinct modules* dedicated to specific functions (*e.g.*, feature extraction, object detection, and interaction recognition), which presents the **challenge ❸**: how to effectively leverage prototypes to optimize the representation space.

To navigate the aforementioned challenges, we propose ProtoHOI, a one-stage HOI detection framework that leverages prototypes to construct a well-structured compositional representation space. For **challenge ❶**, we propose characterizing each HOI triplet by abstracting it through a set of representative prototypes from three conceptual dimensions, *i.e.*, object, verb, and their combinations, which delineate the conceptual boundaries of HOI classes. For **challenge ❷**, we introduce a canidate contruction strategy that selectively chooses proposals within each batch and incorporates a memory bank to facilitate more effective prototype updating. For **challenge ❸**, we implement a contrastive learning paradigm integrating both *fresh* (in current batch) and *stale* (from memory bank) proposals: the former optimizes the interaction decoder, while the latter refines the projection head.

Extensive experiments are conducted on two benchmark datasets, *i.e.*, HICO-DET [13] and V-COCO [14]. Results empirically show that ProtoHOI outperforms state-of-the-art approaches by a solid margin across all compositional zero-shot scenarios in both *Closed-World* and *Open-World* settings. ProtoHOI also delivers competitive performance in fully supervised settings, which further demonstrates the efficacy of the proposed framework for building a robust HOI detector. Moreover, ablation studies verify the strength of our framework design and the effectiveness of our algorithm.

## 2 RELATED WORK

**Human-Object Interaction Detection.** Human-Object Interaction (HOI) is essential for advanced visual understanding, entailing scene and event comprehension, human movements analysis, identification of manipulable objects, and the effect of the human actions on those objects [15]. Detecting HOI requires not only localizing both a human and an object but also inferring their interaction relationship, making it a challenging task [14]. InteractNet [16] first proposes a human-centric approach that constructs HOI triplets by *post-composing* detected humans and objects, establishing the two-stage paradigm for HOI detection [13, 17–20]. Building upon advances in object detection (*e.g.*, DETR [21]), QPIC [22] employs transformer-based encoder-decoder architectures to directly predict *pre-composing* interaction triplets in a set-to-set manner, pioneering the rapid development of the one-stage paradigm [1, 3, 23, 24]. However, these conventional methods are constrained to identifying predefined HOI categories, impairing their generalization to novel interactions.

**Zero-shot HOI Detection.** Zero-shot HOI detection aims to detect both interactions *seen* and *unseen* in the training set, simulating real-world scenarios. Shen et al. [4] first introduce the *unseen combination* setting, which partitions the seen and unseen sets based on HOI triplets. Considering this setting, VCL [25] composes novel interactions with decomposed object and verb features, while ATL [19] leverages affordance-object pairings to discover unseen interactions. More recently, there is a growing interest to enhance zero-shot HOI detection performance by transferring knowledge from pre-trained models. Bansal et al. [5] introduce the *unseen objects* setting and leverage language priors derived from word2vec [26] to mitigate this issue. Liao et al. [1] further proposes the *unseen verb* setting and adopts a distillation framework to align HOI embeddings with CLIP textual representations [27] for this challenge. Most lately, CMMP [28] sets new state-of-the-art results for zero-shot HOI detection by enhancing CLIP representations for human-object pairs.

Despite the commendable prior efforts, our work primarily focuses on the *combinatorial explosion* problem in real-world HOI recognition, which is a critical yet long-overlooked challenge.

**Compositional Learning.** *Compositionality* constitutes a fundamental principle in human visual cognition, denoting the cognitive capacity to represent complex entities through a limited set of simple, reusable components [29]. Early research primarily explores compositional learning in visual question answering [30, 31] and image generation [32, 33]. In the field of visual recognition, Misra et al. [34] introduce the challenge of composing unseen combinations of primitive visual concepts under the zero-shot learning paradigm. Mancini et al. [35] further extend this task to an open-world setting, where the search space comprises numerous unseen compositions, including infeasible ones. This line of research shares parallels HOI recognition in compositional properties, *i.e.*, adjective-noun *vs.*verb-noun pairs, which inspires studies to investigate compositional learning in HOI detection. For instance, Kato et al. [36] use external knowledge graphs and graph convolutional networks to compose classifiers for verb-noun pairs, while Hou et al. [25] separate verbs and objects in the feature space and compose novel interactions through feature stitching. However, *Open-World* zero-shot learning for HOI compositions remains a virgin territory, which treats the lack of constraints in the output space for unseen concepts as a necessity for practical scenarios.

**Prototype Learning.** Prototype-based methods operate by comparing observations or stimuli to a set of reference prototypes or exemplars based on *similarity* [37], which aligns with established concepts in the cognitive psychology and neuroscience. Nearest neighbor rule [38] is one of the most established prototype-based classifiers, which utilizes all training instances as prototypes to assign labels to new samples by measuring their distance. Nearest centroids [39, 40] offers an efficient solution by selecting representative class centroids as prototypes, rather than the entire training dataset. These prototypes provide several compelling virtues, *e.g.*, *structured representation organization* and *transparent decision-making*. These strengths spurr increasing interest in integrating prototype learning with deep learning models across diverse areas, such as supervised learning [11, 41], unsupervised learning [42, 43], few-shot learning [44, 45], and zero-shot learning [46, 47].

Although prototype learning has been extensively studied in fundamental tasks such as classification [48] and semantic segmentation [49], its potential to more complex downstream tasks remains underexplored. In the paper, we focus on adapting prototype learning to HOI detection, leveraging its advantageous properties to enhance the zero-shot performance in *open-world* scenarios.

## 3 METHOD

### 3.1 PROBLEM SETUP

Let $\mathcal{A} = \{a_1, a_2, \ldots, a_M\}$ and $\mathcal{O} = \{o_1, o_2, \ldots, o_N\}$ denote the sets of verb and object, respectively. The Cartesian product $\mathcal{C} = \mathcal{A} \times \mathcal{O}$ defines the complete space of all possible verb-object compositions, while $\mathcal{C}_p \subset \mathcal{C}$ represents the predefined compositions available in the labeled dataset, which is usually sparse, *i.e.*, $|\mathcal{C}_p| \ll |\mathcal{C}|$. The set $\mathcal{C}_p$ is partitioned into a seen $\mathcal{C}_s$ and an unseen $\mathcal{C}_u$ composition set . In the zero-shot setting, given a training set $\mathcal{T} = \{(x_i, c_i) | x_i \in \mathcal{X}, c_i \in \mathcal{C}_s\}_{i=1}^{N}$, the objective is to learn a model $f : \mathcal{X} \to \mathcal{C}_t$ that generalizes to compositions in the test space $\mathcal{C}_t$. Existing works on zero-shot HOI detection [4, 24, 28, 50] assume that the unseen composition set is known a priori, thereby constraining the test space to the predefined set, *i.e.*, $\mathcal{C}_t = \mathcal{C}_p$. However, this assumption often fails in real-world applications, where the test space typically encompasses all compositions, *i.e.*, $\mathcal{C}_t = \mathcal{C}$. In

Figure 2: Overview of our prototype-based learning framework for zero-shot HOI detection.

this paper, we formally discuss these two zero-shot HOI detection scenarios and define the former as the *Closed-World* (CW) setting and the latter as the *Open-World* (OW) setting.

## 3.2 MODEL OVERVIEW

ProtoHOI is a principled framework that can be seamlessly integrated into both one-stage and two-stage architectures. Here, we present a unified HOI detection pipeline that establishes two identification branches for object detection and HOI recognition. Given an input image $I \in \mathbb{R}^{H_0 \times W_0 \times 3}$, a visual encoder is employed to extract the visual features $I \in \mathbb{R}^{H \times W \times 3}$. In the object detection branch, these features are retrieved by an instance decoder $\mathcal{D}_{ins}$ via two distinct sets of queries $Q \in \mathbb{R}^{N \times C}$:

$$\hat{Q} = \mathcal{D}_{ins}(I, Q), \tag{1}$$

where $\hat{Q}$ are further transformed into bounding boxes $B_h$, $B_o$ and object labels $C_o$. In the HOI recognition branch, an interaction decoder $\mathcal{D}_{int}$ is adopted to update the interaction queries $Q_{hoi}$ by leveraging the extracted features $I$:

$$\hat{Q}_{hoi} = \mathcal{D}_{int}(I, Q_{hoi}). \tag{2}$$

Finally, these interaction queries $\hat{Q}_{hoi}$ are fed into the projection head $h$ for the HOI predictions.

## 3.3 HOI DETECTION: FROM CLOSED TO OPEN WORLD

The paradigm shift from closed-world to open-world introduces several significant challenges for HOI detection. **First**, the full combinatorial search space $\mathcal{C}$ is typically prohibitively large for traditional parametric classification heads, leading to excessive computational complexity and resource demands. Alternatively, we project both images and text-based compositions into a shared embedding space and replace the discriminative classifiers by computing cosine similarities between them:

$$f(x) = \arg \max_{c \in \mathcal{C}} \cos(\omega(x), \phi(c)), \tag{3}$$

where the mapping $\omega : \mathcal{X} \to \mathcal{Z}$ projects the image space into the shared embedding space, while $\phi : \mathcal{C} \to \mathcal{Z}$ embeds each HOI composition to the same space using the text encoder of CLIP [27].

**Second**, prior work assumes unseen compositions are known a priori during inference, which artificially limits the number of proposal candidates. However, in the open-world scenario, the number of proposal candidates can become extremely large, and it is ill-suited to take them all for validation, which may introduces a positive bias in within-class precision estimates. Here, we advocate that the HOI community adopt a more rigorous validation protocol by retaining the top-$K$ HOI candidates ($K = 100$) with the highest scores, a strategy already established in one-stage frameworks [1, 22] but not yet adopted in two-stage approaches [28]. To ensure consistency, we recommend applying this constraint in both *Closed-World* and *Open-World* scenarios.

## 3.4 PROTOTYPES AS COMPOSITIONAL DECOUPLER

This section presents a prototype-driven solution to tackle the open-world combinatorial problem. It first outlines the methodology for constructing and updating prototypes, then describes the generation of proposal candidates to support this process, and finally details the prototype-based representation learning strategies alongside its corresponding network training procedure.

**Prototype Modeling.** For each HOI proposal, we construct $K$ prototypes from three perspectives: object, human action, and HOI triplet. This results in a total of $K \cdot (|\mathcal{C}^o| + |\mathcal{C}^a| + |\mathcal{C}^t|)$ prototypes,

where $\mathcal{C}_o$, $\mathcal{C}_a$, and $\mathcal{C}_t$ denote the object categories, action categories, and HOI categories. These prototypes are employed to model intra-class diversity in the candidate set $\mathcal{N} = \{f_i\}_{i=1}^N$, which is derived from the features of HOI proposal. For clarity, we describe only the prototype construction of the HOI triplet; other prototypes (*e.g.*, object and human action) follow an analogous procedure.

Given $N_c$ proposals in the candidate set $\mathcal{N}$ associated with an HOI triplet $c \in \mathcal{C}_t$, our objective is to learn a mapping from these proposals $\{\boldsymbol{f}_i\}_{i=1}^{N_c}$ to the $K$ prototypes $\{\boldsymbol{p}_k^c\}_{k=1}^K$ of $c$. This mapping is formalized as an assignment matrix $\boldsymbol{A}^c = [a_i]_{i=1}^{N_c} \in \{0,1\}^{K \times N_c}$, where $a_i$ is the one-hot assignment vector of proposal $\boldsymbol{f}_i$ over the $K$ prototypes. The matrix $\boldsymbol{A}^c$ can be derived by solving an optimization problem that maximizes the similarity between the proposals and the prototype:

$$\max_{\boldsymbol{A}^c \in \mathcal{A}^c} \text{Tr}(\boldsymbol{A}^{c\top} \boldsymbol{P}^{c\top} \boldsymbol{F}^c) + \varepsilon h(\boldsymbol{A}^c), \tag{4}$$

where $h$ is the entropic constraint, *i.e.*, $h(\boldsymbol{A}^c) = \sum_{i,k} -a_{i,k} \log a_{i,k}$, and $\varepsilon > 0$ governs the smoothness of the mapping. As noted by [51], this optimization generally yields a trivial solution with all proposals collapsing to a single prototype. To prevent this, we constrain the matrix to an element of the transportation polytope [52], formulated as:

$$\mathcal{A}^c = \big\{ \boldsymbol{A}^c \in \mathbb{R}_+^{K \times N_c} | \boldsymbol{A}^c \mathbf{1}^K = \mathbf{1}^{N_c}, \boldsymbol{A}^c \mathbf{1}^{N_c} = \frac{N_c}{K} \mathbf{1}^K \big\}, \tag{5}$$

where $\mathbf{1}^K / \mathbf{1}^{N_c}$ represents the $K/N_c$-dimension all-ones vector. These constraints mitigate proposal-prototype collapsing by ensuring that each proposal is exclusively assigned to a single prototype, and each prototype is selected at least times $N_c/K$ on average. On this basis, the optimization problem in Eq. 4 can be expressed in the form of a normalized exponential matrix [53]:

$$\boldsymbol{A}^{c*} = \text{diag}(\alpha) \exp\big(\frac{\boldsymbol{P}^{c\top} \boldsymbol{F}^c}{\varepsilon}\big) \text{diag}(\beta), \tag{6}$$

where exponentiation is applied element-wise, and $\alpha \in \mathbb{R}^K$ and $\beta \in \mathbb{R}^{N_c}$ denote renormalization vectors computed through a few iterations of the Sinkhorn-Knopp algorithm [53].

Give the non-learnable nature of the prototypes, we adopt an online updating strategy to keep them fresh throughout the network training process:

$$\boldsymbol{p}_k^c \leftarrow \mu \boldsymbol{p}_k^c + (1 - \mu) \bar{\boldsymbol{f}}_k^c, \tag{7}$$

where $\mu \in [0,1]$ is a momentum coefficient, and $\bar{\boldsymbol{f}}_k^c$ denotes the cluster centroid for the $(c,k)$-cluster, computed as the mean of feature vectors from proposal features assigned to this cluster.

**Candidate Contruction.** Unlike classification [48] or segmentation [49, 54] tasks, HOI detection typically generates numerous proposals *containing substantial redundant or irrelevant information*, which pose a significant challenge ❶ for candidate construction. To mitigate ❶, we use employ bipartite matching based on ground truth to select the most reliable proposals as candidates. However, this raises another challenge ❷: since the number of annotated HOI triplets per batch is substantially smaller than the number of prototypes, and HOI categories follow a long-tailed distribution, *relying solely on per-batch proposals as the candidate set may introduce biases due to uneven class-wise update frequencies*. To address ❷, we maintain a memory bank $\mathcal{M}$ that accumulates matched proposal features in each batch and updates prototypes only once it reaches capacity $|\mathcal{M}|$.

**Proposal-Prototype Compositional Contrastive Learning.** We posit that a well-generalizable representation space should not only provide discriminative decision boundaries for distinct HOI categories but also maintain structural integrity to capture diverse intrinsic patterns from involved objects, humans and their combinations. To achieve this, we promote intra-class compactness and inter-class discrimination at the prototype level by introducing a *proposal-prototype compositional contrastive loss*, which operates across three attribute dimensions (*i.e.*, object, human action, and HOI categories) of proposal features. First, we define the contrastive loss based on the assign probability matrix $\boldsymbol{A}^c$, as:

$$\mathcal{L}_{\text{cont}}(\boldsymbol{f}_i, \boldsymbol{P}^c) = -\frac{1}{|\mathcal{M}|} \sum_{i \in \mathcal{M}} \log \frac{\sum_{\boldsymbol{p}^+ \in \mathcal{P}^+} \exp(\boldsymbol{f}_i^\top \boldsymbol{p}^+ / \tau)}{\sum_{\boldsymbol{p}^+ \in \mathcal{P}^+} \exp(\boldsymbol{f}_i^\top \boldsymbol{p}^+ / \tau) + \sum_{\boldsymbol{p}^- \in \mathcal{P}^-} \exp(\boldsymbol{f}_i^\top \boldsymbol{p}^- / \tau)}, \tag{8}$$

where $\mathcal{P}^+$ refers to the prototypes assigned to the proposal feature vector $\boldsymbol{f}_i$, $\mathcal{P}^-$ represents the remaining set of irrelevant prototypes, and $\tau$ is the temperature parameter. To extend such a constrastive setup with our HOI compositions, it is essential to consider a fact: although HOI proposals

may belong to distinct HOI categories, they can share identical objects or human actions, which may lead to certain similar visual patterns. For instance, *human ride horse* and *human ride bicycle* might exhibit similar human poses or spatial configurations, while *human catch frisbee* and *human throw frisbee* may exhibit comparable visual layouts. To account for these hierarchical structures, this final *proposal-prototype compositional contrastive loss* is formulated as:

$$\mathcal{L}_{\text{CCL}}(\boldsymbol{f}_i, \boldsymbol{P}_o^c, \boldsymbol{P}_a^c, \boldsymbol{P}_t^c) = \mathcal{L}_{\text{cont}}(\boldsymbol{f}_i, \boldsymbol{P}_o^c) + \mathcal{L}_{\text{cont}}(\boldsymbol{f}_i, \boldsymbol{P}_a^c) + \mathcal{L}_{\text{cont}}(\boldsymbol{f}_i, \boldsymbol{P}_t^c). \tag{9}$$

**Network Training.** During model training, prototypes in the memory bank originate from different input images and thus cannot be utilized for learning in transformer-based structure, as self/cross-attention mechanisms operate at the image level. To better leverage prototypes for recalibrating network modules in protoHOI, we propose two training strategies: **i)** *stale prototype learning*. Stale, heterologous yet numerous proposal prototypes from the memory bank are leveraged to optimize the linear projection head $h$; **ii)** *fresh prototype learning*. Fresh, homologous but fewer proposal prototypes within each batch are utilized to optimize the interaction decoder $\mathcal{D}_{int}$. This decoupled design enables us to effectively address the complexity of incorporating prototype learning into the HOI detection pipeline, thereby allowing it to benefit from the associated advantages.

## 4 EXPERIMENTS

### 4.1 EXPERIMENT SETTINGS

**Datasets.** We evaluate on HICO-DET [13] and V-COCO [14]. HICO-DET contains 47,776 images (38,118 for training and 9,658 for testing), with 600 HOI classes derived from 80 object categories and 117 actions. V-COCO includes 10,396 images (5,400 for training and 4,964 for testing), with 29 action categories (including 4 body motions), forming 263 HOI classes using 80 object categories.

**Fully-supervised Evaluation.** In line with [13], we use mean average precision (mAP) for model evaluation. For HICO-DET, we report results on two setups: **1)** Default, where mAP is computed over the entire dataset; **2)** Konwn Object, where mAP is computed on a subset containing the object. For V-COCO, we consider two scenarios: **1)** Scenario 1 for all 29 action categories, including the 4 body motions; **2)** Scenario 2 for the 25 action categories, excluding the no-object HOI categories.

**Zero-shot Evaluation.** Zero-shot evaluation is categorized into *Closed-World* (CW) and *Open-World* (OW) settings based on whether HOI compositions are known a priori. On HICO-DET, three evaluation settings are employed: **1)** Unseen Composition (UC), where the training data contains all categories of object and verb but misses some HOI triplet categories; **2)** Rare First Unseen Combination (RF-UC), where rare HOIs are prioritized in the held-out set; **3)** Non-rare First Unseen Combination (NF-UC), where frequent HOIs are held out, resulting in a smaller and more challenging training set. These settings operate within a composition space of 480 seen triplets during training, with 600 possible compositions for CW and 9,360 for OW scenarios during inference.

**Network Architecture.** Two variants of ProtoHOI are developed: a one-stage variant ProtoHOI[†] based on HOICLIP [24] and a two-stage variant ProtoHOI[‡] grounded in CMMP [28], with network architectures adhering to their respective configurations. We set the number of prototypes $K = 4$, the momentum coefficient $\mu = 0.9$, and the memory bank size to 2000.

**Training.** Following prior work [28], ProtoHOI is initialized with DETR fine-tuned on the training set of HICO-DET. The one-stage ProtoHOI[†] is trained for 90 epochs using AdamW with a learning rate of $5 \times 10^{-5}$ and a weight decay of $10^{-4}$, decayed by a factor of 10 every 30 epochs, with a batch size of 6. In contrast, the two-stage ProtoHOI[‡] is trained for 20 epochs with a batch size of 8. It is optimized using AdamW with a learning rate of $1 \times 10^{-3}$ and a weight decay of $10^{-4}$.

**Inference.** We retain top-$K$ proposals ($K = 100$) with the highest confidence scores for final predictions without applying any additional data augmentation at test time.

### 4.2 ZERO-SHOT HOI DETECTION

We evaluate ProtoHOI against with existing HOI detection methods under three compositional zero-shot settings, *i.e.*, UC, RF-UC and NF-UC, in both *Closed-World* and *Open-World* scenarios.

Table 1: **Zero-shot HOI Detection Results** on HICO-DET [13] `test`. UC denotes unseen composition, while RF-UC and NF-UC denote rare first and non-rare first unseen composition, respectively.

|  | Method | UC | | | NF-UC | | | RF-UC | | |
|---|---|---|---|---|---|---|---|---|---|---|
|  |  | Unseen | Seen | Full | Unseen | Seen | Full | Unseen | Seen | Full |
| *CW* | FCL [55][CVPR21] | - | - | - | 18.66 | 19.55 | 19.37 | 13.16 | 24.23 | 22.01 |
|  | ATL [19][CVPR21] | - | - | - | 18.25 | 18.78 | 18.67 | 9.18 | 24.67 | 21.57 |
|  | RLIP [56][NeurIPS22] | - | - | - | 20.27 | 27.67 | 26.19 | 19.19 | 33.35 | 30.52 |
|  | GEN-VLKT [1][CVPR22] | - | - | - | 25.05 | 23.38 | 23.71 | 21.36 | 32.91 | 30.56 |
|  | EoID [57][AAAI23] | 23.01 | 30.39 | 28.91 | 26.77 | 26.66 | 26.69 | 22.04 | 31.39 | 29.52 |
|  | HOICLIP [24][CVPR23] | 23.15 | 31.65 | 29.93 | 26.39 | 28.10 | 27.75 | 25.53 | 34.85 | 32.99 |
|  | CLIP4HOI [50][NeurIPS23] | 27.71 | 33.25 | 32.11 | 31.44 | 28.26 | 32.03 | 28.47 | 35.48 | 34.08 |
|  | CMMP [28][ECCV24] | 29.60 | 32.39 | 31.84 | 32.09 | 29.71 | 30.18 | 29.45 | 32.87 | 32.18 |
|  | ProtoHOI† (Ours) | 28.02 | **36.37** | **34.70** | 31.26 | **33.43** | **33.00** | 29.00 | **36.67** | **35.14** |
|  | ProtoHOI‡ (Ours) | **30.78** | 33.41 | 32.88 | **34.33** | 31.52 | 32.08 | **30.27** | 34.45 | 33.61 |
| *OW* | GEN-VLKT [1][CVPR22] | - | - | - | 20.11 | 18.62 | 18.92 | 17.42 | 23.93 | 22.63 |
|  | HOICLIP [24][CVPR23] | 18.19 | 25.59 | 23.99 | 21.89 | 20.89 | 21.06 | 20.18 | 25.89 | 24.68 |
|  | CMMP [28][ECCV24] | 27.26 | 30.77 | 30.07 | 28.55 | 28.28 | 28.33 | 28.19 | 31.47 | 30.81 |
|  | ProtoHOI† (Ours) | 24.39 | **32.95** | **31.24** | 25.33 | **29.36** | 28.55 | 25.02 | **32.74** | 31.19 |
|  | ProtoHOI‡ (Ours) | **28.58** | 31.82 | 31.17 | **31.03** | 28.12 | **28.70** | **29.07** | 32.55 | **31.85** |

***Closed-World.*** Table 1 illustrates that ProtoHOI† outperforms all one-stage competitors by a solid margin across all zero-shot settings. For instance, under the **UC** settings, it yields gains of 4.87%, 4.72%, and 4.77% mAP on the Full, Rare, and Non-Rare splits, compared with HOICLIP [24]. Compared to CLIP4HOI [50], ProtoHOI† maintains its lead on the Full set, achieving improvements of 2.59 (UC), 0.97% (NF-UC), and 1.07% (RF-UC). As seen, two-stage methods, *e.g.*, CLIP4HOI [50] and CMMP [28], lag behind one-stage approaches on Full and Seen sets, but achieve superior performance on Unseen categories. The two-stage variant ProtoHOI‡ also inherits this characteristic, surpassing the CMMP [28] by 1.18% (UC), 2.24% (NF-UC), and 0.82% (RF-UC) on Unseen set.

***Open-World.*** To rigorously assess the generalization capability of ProtoHOI in real-world settings, we benchmark three state-of-the-art HOI detection methods: two one-stage frameworks (*i.e.*, GEN-VLKT [1] and HOICLIP [24]) and one two-stage frameworks (*i.e.*, CMMP [28]). We extend them to the *Open-World* setting by modifying their classification heads to handle the full combinatorial output space $\mathcal{A} \times \mathcal{O}$. As anticipated, all methods exhibited significant performance degradation, particularly one-stage approaches. For instance, HOICLIP [24] exhibits reductions of 5.97%, 6.69%, and 8.31% on the full sets of UC, NF-UC, and RF-UC, respectively, relative to their prior results in the *Closed-World* setting. In contrast, CMMP [28] exhibits relatively small decrease of approximately 1%-2%. These results demonstrate that open-world zero-shot HOI detection presents a more challenging and realistic task formulation, warranting further investigation. Nonetheless, ProtoHOI still demonstrates superior performance compared to other competitors under this scenario.

### 4.3 FULLY-SUPERVISED HOI DETECTION

This work primarily investigates the zero-shot capabilities of HOI detectors. Nevertheless, to fully substantiate ProtoHOI's efficacy, we also evaluate its performance under a fully-supervised setting on HICO-DET [13] and V-COCO [14]. As shown in Table 2, ProtoHOI establishes new state-of-the-art results on HICO-DET under both *Default* and *Known Object* settings. In particular, it surpasses the previous state-of-the-art method Pose-Aware by 0.47%, 0.53% and 0.46% mAP on the Full, Rare, and Non-Rare settings. Moreover, ProtoHOI achieves a more significant performance gain over the zero-shot detector CMMP [28], with an improvement of 4.07% mAP under the Full setting. For V-COCO, ProtoHOI observes consistent improvements over prior methods, surpassing the state-of-the-art method STIP [20] in Scenario 1. Notably, our approach significantly surpasses existing zero-shot detectors, outperforming HOICLIP [24] by 1.9% and CMMP [28] by 5.6% on Scenario 2.

### 4.4 DIAGNOSTIC EXPERIMENTS

As shown in Table 3, we conduct ablation studies on HICO-DET in zero-shot NF-UC settings.

Table 2: **Fully-supervised HOI Detection Results** on the HICO-DET [13] and V-COCO [14] test.

| Method | HICO-DET (DF) | | | HICO-DET (KO) | | | V-COCO | |
|---|---|---|---|---|---|---|---|---|
| | Full | Rare | Non-rare | Full | Rare | Non-rare | $AP_{role}^{S1}$ | $AP_{role}^{S2}$ |
| VCL [25][ECCV20] | 19.43 | 16.55 | 20.29 | 22.00 | 19.09 | 22.87 | 48.3 | - |
| PPDM [58][CVPR20] | 21.73 | 13.78 | 24.10 | 24.58 | 16.65 | 26.84 | - | - |
| QPIC [22][CVPR21] | 29.07 | 21.85 | 31.23 | 31.68 | 24.14 | 33.93 | 58.8 | 61.0 |
| HOTR [59][CVPR21] | 23.46 | 16.21 | 25.60 | - | - | - | 55.2 | 64.4 |
| CDN [23][NeurIPS21] | 31.44 | 27.39 | 32.64 | 34.09 | 29.63 | 35.42 | 61.2 | 63.8 |
| STIP [20][CVPR22] | 32.22 | 28.15 | 33.43 | 35.29 | 31.43 | 36.45 | 65.1 | **69.7** |
| DOQ [60][CVPR22] | 33.28 | 29.19 | 34.50 | - | - | - | 63.5 | - |
| GEN-VLKT [1][CVPR22] | 33.75 | 29.25 | 35.10 | 37.80 | 34.76 | 38.71 | 62.4 | 64.4 |
| ADA-CM [2][ICCV23] | 33.80 | 31.72 | 34.42 | - | - | - | 56.1 | 61.5 |
| HOICLIP [24][CVPR23] | 34.59 | 31.12 | 35.74 | 37.61 | 34.47 | 38.54 | 63.5 | 64.8 |
| PViC [61][ICCV23] | 34.69 | 32.14 | 35.45 | 38.14 | 35.38 | 38.97 | 62.8 | 67.8 |
| Pose-Aware [62][CVPR24] | 35.86 | 32.48 | 36.86 | **39.48** | 36.10 | **40.49** | 61.1 | 66.6 |
| CMMP [28][ECCV24] | 32.26 | 33.53 | 33.24 | - | - | - | - | 61.2 |
| ProtoHOI† (Ours) | **36.33** | **33.01** | **37.32** | 39.39 | **37.03** | 40.09 | **65.4** | 66.8 |

Table 3: **A Set of Ablative Experiments** of ProtoHOI† on HICO-DET [13] test in CW setting.

| $K$ | NF-UC | | |
|---|---|---|---|
| | Unseen | Seen | Full |
| 1 | 30.85 | 32.26 | 31.98 |
| 2 | 30.92 | 32.32 | 32.04 |
| 3 | 30.76 | 32.72 | 32.33 |
| 4 | **31.26** | **33.43** | **33.00** |
| 5 | 30.49 | 32.58 | 32.16 |

(a) Prototype Number ($K$)

| $|\mathcal{M}|$ | NF-UC | | |
|---|---|---|---|
| | Unseen | Seen | Full |
| 0 | 29.93 | 32.55 | 32.02 |
| 500 | 30.42 | 32.85 | 32.37 |
| 1000 | 31.01 | 32.97 | 32.58 |
| 2000 | **31.26** | **33.43** | **33.00** |
| 4000 | 30.25 | 32.95 | 32.42 |

(b) Memory Bank Size ($|\mathcal{M}|$)

| $\mu$ | NF-UC | | |
|---|---|---|---|
| | Unseen | Seen | Full |
| 0 | 30.52 | 32.14 | 31.82 |
| 0.8 | 31.13 | 32.69 | 32.38 |
| 0.9 | **31.26** | **33.43** | **33.00** |
| 0.99 | 31.07 | 33.10 | 32.69 |
| 0.999 | 30.89 | 32.71 | 32.36 |

(c) Momentum Coefficient ($\mu$)

**Prototype Number $K$.** Table 3a investigates the effect of the number of prototypes per HOI category. For $K = 1$, each prototype is defined as the mean embedding of all proposal features stored in the memory bank that belong to the corresponding category. This baseline achieves mAP of 30.85%, 32.26%, and 31.98% on the unseen, seen, and full splits. The performance of ProtoHOI consistently improves as the number of prototypes per HOI category increases, peaking at $K = 4$, *i.e.*, unseen: 31.26%, seen: 33.43%, full: 33.00%. This observation indicates that modeling intra-class diversity via multiple prototypes can enhance the zero-shot performance of ProtoHOI. However, increasing $K$ beyond 4 leads to performance deterioration, likely due to excessive clustering produces some trivial prototypes, which may compromise the effectiveness of other prototypes.

**Memory Bank Size $|\mathcal{M}|$.** Table 3b examine the effect of the size of memory bank. As seen, ProtoHOI consistently benefits from larger memory sizes, showing gradual improvements in performance. However, when $|\mathcal{M}| > 2000$, further increasing the memory size can even lead to negative effects. This may be attributed to the fact that larger memory sizes tend to accumulate stale proposals, thereby introducing inconsistencies in the representation that impair prototype modeling.

**Momentum Coefficient $\mu$.** Table 3c analyzes the role of the momentum coefficient $\mu$, which governs the update rate of prototypes. Our results demonstrate that ProtoHOI achieves optimal performance with a relatively large update magnitude, *i.e.*, $\mu = 0.9$. This can be attributed to the inherent complexity of the HOI detection, which involves the simultaneous optimization of multiple loss functions, leading to less stable training dynamics. In contrast, setting $\mu = 0$, *i.e.*, completely discarding the previous prototypes, leads to a significant performance drop.

**Model Efficiency Analysis.** Table 4 benchmarks the model efficiency of ProtoHOI against prior methods, reporting the number of parameters, FLOPs, and FPS on an NVIDIA RTX 4090 GPU. ProtoHOI†, with a similar network architecture to HOICLIP, achieves better performance with

Table 4: Comparison of parameters and running efficiency.

| Method | Params(M) | FLOPs(G) | FPS |
|---|---|---|---|
| PPDM [58] [CVPR20] | 194.9 | 121.63 | 15.28 |
| GEN-VLKT [1] [CVPR22] | 42.53 | 85.64 | 22.36 |
| HOICLIP [24] [CVPR23] | 67.12 | 89.22 | 34.98 |
| CMMP [28] [ECCV24] | 193.43 | 106.89 | 21.96 |
| ProtoHOI† (Ours) | 64.47 | 86.29 | 19.21 |

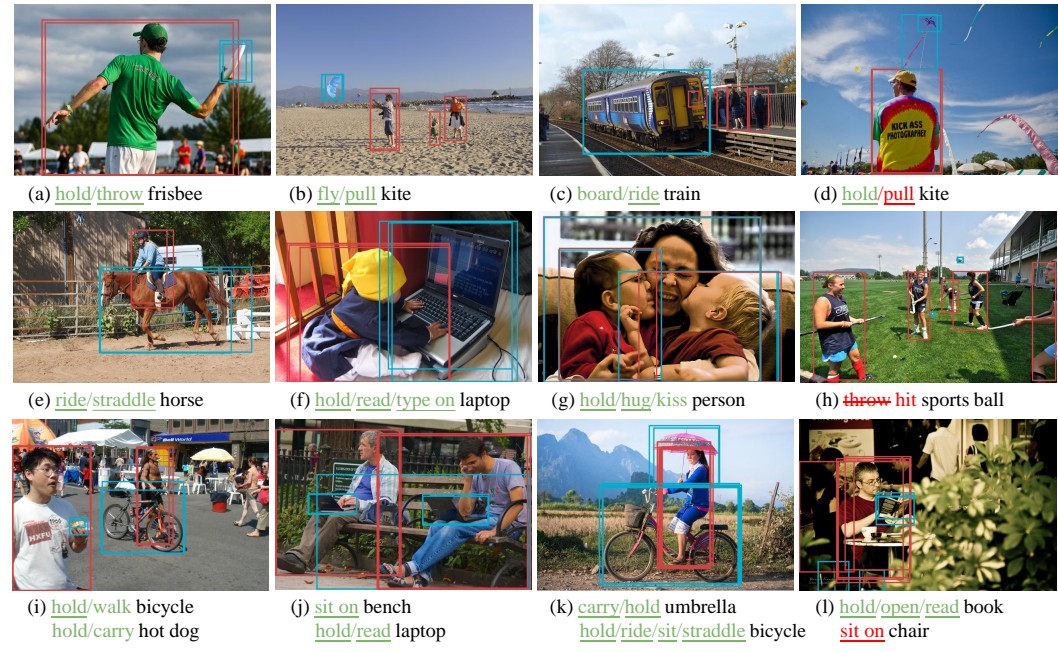

Figure 3: Visualization of ProtoHOI[†] results on HICO-DET [13] test under the NF-UC setting. Detected interaction are marked in Green, missed interactions and objects are highlighted in Red, while novel combinations are underlined.

fewer parameters and lower computational cost (FLOPs), e.g., 1.74 % mAP improvement in the fully supervised setting and 2.25 % for NF-UC for closed-world zero-shot setting.

## 4.5 QUALITATIVE RESULTS

Figure 3 presents the visualization results on the HICO-DET test set. As seen, our method effectively detects human-object interactions across diverse scenarios. For instance, in (k), our method detects all potential actions, while in (c), it successfully distinguishes interactions involving individuals and the train, *i.e.*, identifying *ride* inside the train and *board* adjacent to it. Additionally, we present several failure cases. Our method fails to discern interactions in severely occluded scenes, such as the overlooked chair-human interaction in the bottom-left of (l). It also struggles with ambiguous intentions when viewpoint is limited, as shown in (d), where the human's back view makes it difficult to confirm a *pull* action. Moreover, our method misclassified the airborne rugby ball in (h) as *throw*, but it is conventionally associated with the action *hit*. These results highlight persistent challenges in HOI detection, indicating a need for further investigation.

## 5 CONCLUSION

In this work, we emphasize the compositional generalization capability in zero-shot HOI detection by introducing a new benchmark that extends the problem from the closed world to the open world scenario, *i.e.*, without restricting the search space of possible HOI combinations. Empirical experiments show that current state-of-the-arts exhibit significant performance degradation on unseen combination under the open-world scenario. To open this avenue, we devise ProtoHOI, a one-stage HOI detection framework that leverages prototypes to construct a structured representation space for facilitating compositional learning via: **i)** a compositional prototype construction and update mechanism is designed to distill the inherent properties of objects (*e.g.*, affordance, functionality) and actions (*e.g.*, human poses, gestures) involved in HOI from proposal representations. **ii)** *fresh* and *stale* proposal-prototype contrastive learning strategies are employed to facilitate modular optimization of its components. Evaluations on two benchmark datasets in diverse zero-shot settings demonstrate the efficacy of ProtoHOI, with ablations validating the significance of its core designs.

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

## SUMMARY OF THE APPENDIX

This appendix contains additional details for the ICLR 2026 submission, titled *"Composing Human-Object Interaction with Decoupled Prototype for Zero-shot Learning"*, and is organized as follows:

- §A discusses our limitations.
- §B discusses our broader impact.
- §C discusses our directions of our future work.
- §D provides the Large Language Models (LLMs) usage statement in this work.

## A    LIMITATION

One limitation of our approach is the introduction of additional prototype construction and update operations during training, which increases the computational overhead. However, during inference, our method does not incur any additional computational burden or resource consumption.

## B    SOCIAL IMPACT

This work investigates HOI detection in open-world scenarios. However, inaccurate predictions in real-world applications (*e.g.*, autonomous driving and human-robot interaction) may compromise human safety. To mitigate this risk, we recommend implementing a security protocol addressing algorithm malfunctions during practical deployments.

## C    FUTURE WORK

In future work, we aim to explore more effective integration of prototype learning into the HOI detection pipeline to enhance the model's performance on open-world tasks. Additionally, building upon compositional challenges as the core focus, we will further investigate applications in broader domains of visual relationship understanding, such as scene graph generation.

## D    LLM USAGE STATEMENT

We employed large language models (LLMs) as auxiliary tools during manuscript preparation. Their use was strictly limited to surface-level editing tasks, including grammar correction, minor rephrasing and stylistic improvements to enhance readability. At no point did we rely on LLMs for generating research ideas, methods, experiments, or conclusions. All technical content and analysis presented in this paper are the sole work of the authors.

