# OpenReview forum: "Composing Human Object Interaction with Decoupled Prototype for Zero-shot Learning"
_ICLR.cc/2026/Conference — ICLR 2026 Conference Withdrawn Submission_

### Official Review · Reviewer_k2yr · 2025-10-25

**Soundness:** 3
**Presentation:** 2
**Contribution:** 2
**Rating:** 4
**Confidence:** 4

**Summary:**

The paper proposes a prototype-based framework for HOI detection, aiming to address challenges of intra-class diversity, redundant proposals, and representation optimization. While the method appears novel and achieves competitive results, its connection to the stated challenges is unclear, and the claimed ability to handle arbitrary action–object combinations lacks explicit justification. Moreover, the evaluation is limited to HICO-DET, which includes only predefined HOIs, leaving the method’s effectiveness for open-world compositional scenarios unverified.

**Strengths:**

1. The paper focuses on a critical open problem, numerous action and object combinations in HOI.
2. The proposed challenges is meaningful, including large class diversity, noisy proposals and representation optimization.
3. The proposed prototype-based representation learning seems to be novel and interesting.

**Weaknesses:**

1. The paper does not include a related paper [a], which addresses the same problem of limited predefined HOI label space and adopts an identical compositional setting (seen actions and objects but unseen combinations). The lack of discussion on this closely related work weakens the positioning and novelty of the paper. This paper should also be a baseline for comparison.
2. The proposed solution to challenge 1 (using three separate feature representations for object, verb, and their combination) is not novel, as similar decompositional strategies have already been explored in prior works [b].
3. The solution to challenge 2 mainly relies on standard bipartite matching (Hungarian algorithm) to reduce redundant proposals and a memory bank to enlarge the batch for prototype updates. However, the method does not explicitly control that all HOI classes are included in the batch. For long-tailed or rare classes, the memory bank cannot ensure balanced updates since some classes may still have few or no samples. Therefore, the proposed strategy cannot effectively address class-level imbalance as claimed.
4. All experiments are conducted only on HICO-DET. If the paper aims to address the compositional or open-world combination problem, it should also evaluate on a benchmark designed for open-compositional HOI understanding, such as Swig-HOI [c].
5. Additionally, it remains ambiguous how the method can handle arbitrary verb–object combinations in the claimed open-world setting, as no explicit compositional mechanism is described.
6. The comparisons in Tables 1 and 2 do not include several recent state-of-the-art zero-shot HOI methods [d, e], whose codes are publicly available and can be evaluated even under the open-world setting. The latest compared method is CMMP (ECCV 2024), making the evaluation incomplete. Although achieving SOTA performance is not the solely important thing for one method, it is good to have a comprehensive comparison.

References:

[a] Discovering Human-Object Interaction Concepts via Self-Compositional Learning, ECCV 2022

[b] Efficient Adaptive Human-Object Interaction Detection with Concept-Guided Memory, ICCV 2023

[c] Learning transferable human-object interaction detectors with natural language supervision, CVPR22

[d] EZ-HOI: VLM Adaptation via Guided Prompt Learning for Zero-Shot HOI Detection, NeurIPS 2024

[e] Locality-Aware Zero-Shot Human-Object Interaction Detection, CVPR 2025

**Questions:**

In summary, although the prototype-based framework appears novel and achieves competitive results, its design lacks a clear connection to the identified challenges. Meanwhile, the solutions mentioned in the paper to address those mentioned challenges, are not novel. These are my key concerns.

---

### Official Review · Reviewer_VYFg · 2025-10-30

**Soundness:** 2
**Presentation:** 2
**Contribution:** 2
**Rating:** 4
**Confidence:** 4

**Summary:**

The authors proposed ProtoHOI as a prototype-based framework for zero-shot HOI detection. It first distills a set of prototypes, then calibrates these prototypes. Extensive experiments and positive results are demonstrated.

**Strengths:**

- The proposed framework appears to be reasonable and effective. By exploiting the powerful compositional generalization ability with the prototype-based framework ProtoHOI, the closed-world zero-shot setting could be extended to an open-world zero-shot setting naturally.

- The results are impressive. Higher mAPs are provided for most experiments compared with the baselines. Qualitative results also shown the efficacy of the proposed method.

**Weaknesses:**

- The performance on both fully-supervised and zero-shot HOI detections is far from swin-Large based SOTA methods [1-3]. If possible, adapting ProtoHOI with swin-large and conducting more comprehensive comparisons would be preferred.

- In Table 1, CMMP is worse than ProtoHOI$^\ddagger$ and better than ProtoHOI$^\ddagger$. However, in Table 4, only ProtoHOI$^\ddagger$ is compared in parameters. It seems CMMP is superior in both computing speed and performance compared to ProtoHOI$^\ddagger$.

- Given the claim on zero-shot HOI detection, it would be more convincing if the proposed method could be compared with SOTA VLMs, both open-sourced and closed-sourced. Though they were not specifically designed for HOI, their grounded vision-language understanding capabilities could be a strong baseline for open-world practice. Also, to better support the contribution of ProtoHOI, it would be more convincing if the necessity of methodologies specifically designed for HOI could be proved by revealing the inherent incapabilities of SOTA VLMs for HOI detection.

[1] Guo Y, Liu Y, Li J, et al. Unseen no more: Unlocking the potential of clip for generative zero-shot hoi detection[C]//Proceedings of the 32nd ACM International Conference on Multimedia. 2024: 1711-1720.

[2] Yuan H, Zhang S, Wang X, et al. Rlipv2: Fast scaling of relational language-image pre-training[C]//Proceedings of the IEEE/CVF international conference on computer vision. 2023: 21649-21661.

[3] Wu E Z Y, Li Y, Wang Y, et al. Exploring pose-aware human-object interaction via hybrid learning[C]//Proceedings of the IEEE/CVF Conference on Computer Vision and Pattern Recognition. 2024: 17815-17825.

**Questions:**

- Please refer to the weakness part.

- Could ProtoHOI be extended to datasets other than HICO-DET? If possible, it would support the feasibility of ProtoHOI for open-world applications.

---

### Official Review · Reviewer_BCaZ · 2025-11-01

**Soundness:** 2
**Presentation:** 2
**Contribution:** 3
**Rating:** 4
**Confidence:** 5

**Summary:**

This paper tackles the problem of zero-shot Human-Object Interaction detection, focusing on the challenge of combinatorial explosion.
They first discuss a zero-shot settin with combinatorial explosion.

Then, they pose ProtoHOI, a framework that learns decoupled prototypes for objects, actions, and full HOI triplets. The goal is to capture the inherent properties of these components to improve compositional generalization.

The method works by distilling prototypes from proposals, maintaining a memory bank, and using a "decoupled" training strategy where stale prototypes (from the memory bank) optimize the projection head and fresh prototypes (from the current batch) optimize the interaction decoder.

The authors claim that ProtoHOI achieves new SOTA results on HICO-DET and V-COCO.

**Strengths:**

1. The open-world setting tackling the combinatorial explosion is well-motivated and practically relevant.
2. The experimental results are comprehensive. The proposed ProtoHOI achieves good performance across different tested scenarios.
3. The authors show their method's effectiveness by integrating it into both a one-stage (HOICLIP-based) and a two-stage (CMMP-based) detector to verify it is applicable to different frameworks of HOI detection.
4. The paper is generally easy to follow and the language usage is satisfying. The motivation to use prototype-based method is clear. The challenges to solve are well listed in the introduction.
5. The authors promise to release the code, which can be helpful for reproducibility.

**Weaknesses:**

1. The contribution of a new setting (or not?) is not clear, not well elaborated, and not distinctive enough compared with existing setting.
    - From the abstract, it seems the author proposes a new task. However, based on the introduction, the difference between the proposed OW setting and existing CW zero-shot setting is if the unseen categories are predefined  or not, which seems not distinctive enough to be a new task.
    - Moreover, after the introduction, the authors basically do not discuss the OW setting at all, except mentioning they verify some existing methods in 4.2.
    - Besides, based on Sec 4.2, we can see some existing methods are also capable of this setting, with comparable performance (CMMP 27.26 vs. ProtoHOI 28.58) to the proposed method.
- Thus, it is not clear whether the authors claim this setting as a new contribution or not.
2. For the 3rd challenge of the three challenges, the authors propose a contrastive learning paradigm, which should be a major contribution. However, it is only briefly mentioned in Sec. 3.4 and no details are given.
4. The authors state that stale prototypes optimize the projection head ($h$) while fresh prototypes optimize the interaction decoder ($\mathcal{D}_{int}$). This specific design choice feels arbitrary and lacks a strong theoretical or intuitive justification. Why this specific design? What is the principle that motivates this "decoupled design"? This is not explained.
5. Multiple closely related previous works are missing. For example, [A, B] discuss the compositional zero-shot learning in HOI detection, which should not be missing as there are only  several previous works focusing on the same combinational explosion issue this paper investigates; [C] focuses on decomposing the representation learning of object, action and human, same as the target of this paper; [D, E] investigate prototype learning for HOI detection and Scene Graph detection (extremely similar to HOI).
6. Missing ablation. The paper provides no ablation study for the "decoupled" training strategy (stale vs. fresh). This is a major omission. The ablations in Table 3 validate the number of prototypes, the memory bank size, and the momentum, but not the core design of how these prototypes are used. Without this ablation, it is impossible to determine if this proposed strategy is actually contributing to the performance, or if the gains simply come from applying a standard prototype-based contrastive loss.
7, The "compositional" aspect of the method feels overstated. The "proposal-prototype compositional contrastive loss" (Equation 9) is simply a sum of three independent contrastive losses—one for object prototypes, one for action prototypes, and one for triplet prototypes. It is not clear how this simple summation enforces any true compositional structure or decoupling.
8. The claim of SOTA performance is not true. The authors are missing a significant number of SOTA methods, either fully-supervised or zero-shot ones, like RLIPv2 (ICCV 23), RelationLMM (TPAMI 25), Discovering Syntactic Cues (CVPR 24).

A. Discovering Human-Object Interaction Concepts via Self-Compositional Learning, ECCV 2022
B. Improving human-object interaction detection via phrase learning and label composition, AAAI 2022.
C. HOI Analysis: Integrating and Decomposing Human-Object Interaction, NeurIPS 2020.
D. Category query learning for human-object interaction classification, CVPR 2023.
E. Prototype-based Embedding Network for Scene Graph Generation, CVPR 2023.

**Questions:**

Please see the weaknesses above. Based on the weaknesses above, I would recommend the authors to revise the manuscript significantly, enriching the contribution clarification (sec 1), the missing related works (sec 2) and compared SOTA methods (sec 4), adding missing method elaboration (sec 3.4), and supplementing with more ablation studies (sec 4). As the weaknesses comments above are already very comprehensively written and took a lot of time, I think it may be not necessary to repeat them again here.

---

### Official Review · Reviewer_9Per · 2025-11-01

**Soundness:** 3
**Presentation:** 3
**Contribution:** 2
**Rating:** 4
**Confidence:** 2

**Summary:**

This paper proposes ProtoHOI, a prototype-based framework for zero-shot Human–Object Interaction (HOI) detection. The key idea is to decouple compositional learning of actions and objects by constructing prototypes and applying a compositional contrastive loss. The authors further introduce an Open-World zero-shot HOI setting, removing the constraint that unseen verb–object compositions are known a priori. Experiments on HICO-DET and V-COCO show improvements over recent CLIP-based methods such as HOICLIP and CMMP.

**Strengths:**

1. This paper clearly highlights a realistic limitation of current HOI detection (Closed-World assumption) and formally proposes the more challenging Open-World (OW) setting.

2. Decoupled prototype learning (object/action/triplet) is well integrated with transformer-based detectors.

3. Comprehensive Evaluation. The experimental evaluation is thorough, including ablations for key hyperparameters (K, μ, |M|).  Implementation details and efficiency analysis are reported.

**Weaknesses:**

1. Section 3.3 lacks substance: The "challenge" of moving to Open-World is addressed by standard practices (projecting into a CLIP space, applying top-K filtering). This is a necessary implementation step, not a conceptual contribution.

2. The paper lacks a theoretical argument for why its specific form of decoupled prototype learning (Eq. 9)  improves compositional generalization. The motivation from human cognition is intuitive but not deeply connected to the mechanics of the representation learning.

3. Qualitative results do not convincingly show how prototypes reshape the representation space.

**Questions:**

1. How does ProtoHOI perform when the CLIP text encoder is replaced by other vision–language backbones (e.g., ALIGN, SigLIP)?

2. In Open-World evaluation, does the model ever produce implausible combinations (e.g., *read horse*)? If so, how are these handled?

3. The authors state that "stale" prototypes (from the memory bank) cannot be used to update the interaction decoder $D_{int}$ due to the nature of attention mechanisms.

    (a) Does this decoupled strategy—using "fresh" prototypes for $D_{int}$ and "stale" prototypes for the projection head $h$—lead to an undesirable discrepancy in the representations seen by these two components?

    (b) Can the authors provide an ablation study to prove this design is necessary? For instance, what is the performance if "fresh prototype learning" is disabled (i.e., $D_{int}$ is not optimized via prototype loss at all)?

4. The compositional contrastive loss in Eq. (9) is a simple sum of three independent losses ($L_o, L_a, L_t$). How does this formulation explicitly encourage the composition of verb and object prototypes to form a novel triplet representation? It seems to only enforce intra-class compactness at three separate levels, rather than modeling their compositional relationship.

---

### Note · Authors · 2025-11-14

I have read and agree with the venue's withdrawal policy on behalf of myself and my co-authors.